# Differences in medical costs for end-of-life patients receiving traditional care and those receiving hospice care: A retrospective study

Ya-Ting Huang[1,2]*, Ying-Wei Wang[3], Chou-Wen Chi[4,5], Wen-Yu Hu[6], Rung Lin, Jr[7,8,9], Chih-Chung Shiao 🄳[2,10], Woung-Ru Tang 🄳[11]*

**1** Department of Nursing, Camillian Saint Mary's Hospital Luodong, Luodong, Yilan, Taiwan, R.O.C, **2** Saint Mary's Junior College of Medicine, Nursing and Management, Sanxing Township, Taiwan, R.O.C, **3** Health Promotion Administration, Ministry of Health and Welfare. Datong Dist., Taipei City, Taiwan, R.O.C, **4** Division of Hematology-Oncology, Chang Gung Memorial Hospital at Linkou, Guishan Dist., Taoyuan City, Taiwan, R.O.C, **5** College of Medicine, Chang Gung University, Kwei-Shan, Tao-Yuan, Taiwan, ROC, **6** Department of Nursing College of Medicine, National Taiwan University, Taipei, Taiwan R.O.C, **7** Department of Anesthesiology, Chang Gung Memorial Hospital at Linkou, Guishan Dist., Taoyuan City, Taiwan, R.O.C, **8** Clinical Informatics and Medical Statistics Research Center, Chang Gung University, Kwei-Shan, Tao-Yuan, Taiwan, ROC, **9** Graduate Institute of Clinical Medicine, Chang Gung University, Kwei-Shan, Tao-Yuan, Taiwan, ROC, **10** Division of Nephrology, Department of Internal Medicine, Camillian Saint Mary's Hospital Luodong, Luodong, Yilan, Taiwan, R.O.C, **11** School of Nursing, Chang Gung University, Kwei-Shan, Tao-Yuan, Taiwan, ROC

* frankie7451@gmail.com(YTH); wtang@mail.cgu.edu.tw(WRT)

**Data Availability Statement:** All relevant data are within the paper and its Supporting Information files. The minimal data set can replicate the study

## Abstract

### Background

Hospice care has a positive effect on medical costs. The correlation between survival time after receiving hospice care and medical costs has not been previously investigated in the literature on Taiwan. This study aimed to compare the differences in medical costs between traditional care and hospice care among end-of-life patients with cancer.

### Methods

Data from Taiwan's National Health Insurance program on all patients who had passed away between 2010 and 2013 were used. Those whose year of death was between 2010 and 2013 were defined as end-of-life patients. The patients were divided into two groups: traditional care and hospice care. We then analyzed the differences in end-of-life medical cost between the two groups.

### Results

From 2010 to 2013, the proportion of patients receiving hospice care significantly increased from 22.2% to 41.30%. In the hospice group, compared with the traditional group, the proportions of hospital stays over 14 days and deaths in a hospital were significantly higher, but the proportions of outpatient clinic visits; emergency room admissions; intensive care unit admissions; use of ventilator; use of cardiopulmonary resuscitation; and use of hemodialysis, surgery, and chemotherapy were significantly lower. Total medical costs were

findings reported, as well as related metadata and methods.

**Funding:** The research was supported by a grant from Camillian Saint Mary's Hospital Research Fund (SMHRF-104004) to C.C. Shiao. The study was based in part on data from the National Health Insurance Research Database (NHIRD) supported by the Bureau of National Health Insurance, Department of Health, and managed by the National Health Research Institutes (NHIRD-104-157). The funders had no role in study design, data collection and analysis, decision to publish, or preparation of the manuscript.

**Competing interests:** The authors have declared that no competing interests exist.

significantly lower. A greater number of days of survival for end-of-life patients when receiving hospice care results in higher saved medical costs.

## Conclusion

Hospice care can effectively save a large amount of end-of-life medical costs, and more medical costs are saved when patients are referred to hospice care earlier.

## Introduction

Because the average life expectancy of the global population has greatly increased and the global population is aging, chronic diseases have become the most common health problem in the world today [1, 2]. When chronic diseases progress to the end of the disease or severe complications occur due to acute deterioration caused by poor disease control, the threat of death increases accordingly [3, 4]. When patients face the threat of death due to disease, hospice care is designed to assist them in achieving a natural death [5, 6].

Taiwan's hospice care has been developing for nearly 30 years, since 1983. Before 2011, the service of hospice care primarily consisted of inpatient hospice care, and secondly home-based hospice care. However, the utilization rate of hospice care in Taiwan was only about 20% with a slow growth rate [7, 8]. The slow growth of hospice care in Taiwan may be explained by differences in cultural customs between different countries [9–11] and restrictions on the patient's "right to know" by family members [9–16]. Lack of availability of hospice care also needs to be considered. Due to the limited number of hospice service institutions and the shortage of hospice beds, terminally ill cancer patients cannot receive hospice care as they wish and must stay in acute medical institutions to continue receiving active and curative treatment [17]. Therefore, National Health Insurance (NHI) developed the Hospice Shared-care Program to improve the rate of hospice care use in Taiwan since 2011, to enable patients who need hospice care to be cared for by hospice care teams in acute wards instead of limiting care to hospice wards. It was hoped that through discussions with hospice teams, the correct knowledge of hospice care could be obtained, and the utilization rate of hospice could be increased [18–20].

Hospice care not only improves the quality of end-stage care for patients and their families [21, 22] but also has a positive effect on medical costs [23–26]. Previous studies have shown that patients receiving hospice care can save US$6,766 to US$7,097 per person compared with those receiving traditional care [27–29].

Previous Taiwanese studies have examined medical expenses in hospice care. In studies on hospice care, basic attributes of patients (e.g., age, sex, residence, and coexisting diseases) are mostly used for matching [23, 25, 30, 31]. Few studies have matched patients according to the number of survival days after hospice care or traditional care, except for those that have examined the basic attributes of patients and coexisting diseases. Domestic and foreign studies on medical expenses in hospice care mainly compare differences in medical expenses in a fixed period of time (e.g., one month or one year before death) among patients at the end of their lives who do or do not receive hospice care [7, 32, 33]. These studies rarely compare the differences in medical expenses between traditional and hospice care using the same number of survival days. This study compared the differences in medical expenses between traditional care patients and terminally ill cancer patients receiving hospice care using the same number of survival days to specifically evaluate the cost-saving of hospice care and to provide a reference for all stakeholders.

## Materials and methods

### Research design and sample selection

The data source for this study was the NHI Database. The file was the 2010 underwriting data file provided by the NHI Administration. Data of 27,378,403 people who were insured from January 1, 2010, to December 31, 2010, were used as the data parent file. One million people were randomly selected from the parent population [34]. Then the death dates of the patients were selected from a sample of 1 million people in 2010, and the end-of-life was defined by dates of death backtracking to the previous year from 2010–2013. Such cases were set as end-stage cases. Next, the health insurance data of cancer patients were separated using the *International Classification of Diseases* (ICD-9-CM code 140–208). If the end-stage patients received more than one hospice care service (whether inpatient hospice care, home-based hospice care, or hospice-shared care) in the year before death, they were placed in the hospice group. If the patients had not received any type of hospice care in the year before death, they were placed in the traditional group.

### Sample matching

After screening out the cases in the hospice group, the number of survival days was calculated by subtracting the first date of receiving inpatient hospice care, home-based hospice care, or hospice-shared care from the date of death. However, hospice-shared care was included as consultation data in the health insurance database, and there was no exact consultation date. The survival days of terminal cancer patients were 7–10 days (39–51%) after receiving hospice-shared care [35, 36]. To minimize underestimation or overestimation of the medical expenses, this study used the discharge date of receiving hospice-shared care to calculate the number of survival days first, and then used the number of hospitalization days of receiving hospice-shared care to calculate the last 10-day interval of the current hospitalization into the number of survival days (10 days for patients with more than 10 days and the original number of days for patients with fewer than 10 days) [37]. Survival days of the traditional group were calculated using the date of the visit or hospitalization and the date of death.

Next, age, gender, income, occupation, Charlson Comorbidity Index (CCI) score, hospital characteristics (medical region such as north, south, central, or east area, hospital level, number of hospital beds, and number of hospice beds) and hospice bed density were treated as the control variables, and whether to accept hospice palliative care or not was treated as the dependent variable. The propensity score was calculated using logistic regression [7, 8]. Data of patients with terminal-stage cancer were first cut in the year of death and then cut for the second time according to the cancer diagnosis. The matching method required one-to-one matching in the same year of death, cancer diagnosis, and survival days with the nearest neighbor matching technique using the propensity score [7, 32, 33, 38].

After matching, cases were classified according to the number of days of survival: 1–3 days, 4–7 days, 8–14 days, 15–30 days, 31–60 days, 61–90 days, 91–180 days, and 180–365 days [39]. Basic patient data, medical treatment data, and related medical expenses were obtained using outpatient and inpatient health insurance records for subsequent analysis. The sample selection and matching flow chart are shown in Fig 1.

### Data collection and ethical considerations

Accumulated Charlson Comorbidity Index (CCI) values for the primary diagnosis and secondary diagnosis codes for outpatient and inpatient services in the year before death were calculated [40]. Characteristics of the hospital were found mainly by looking at the hospitals with

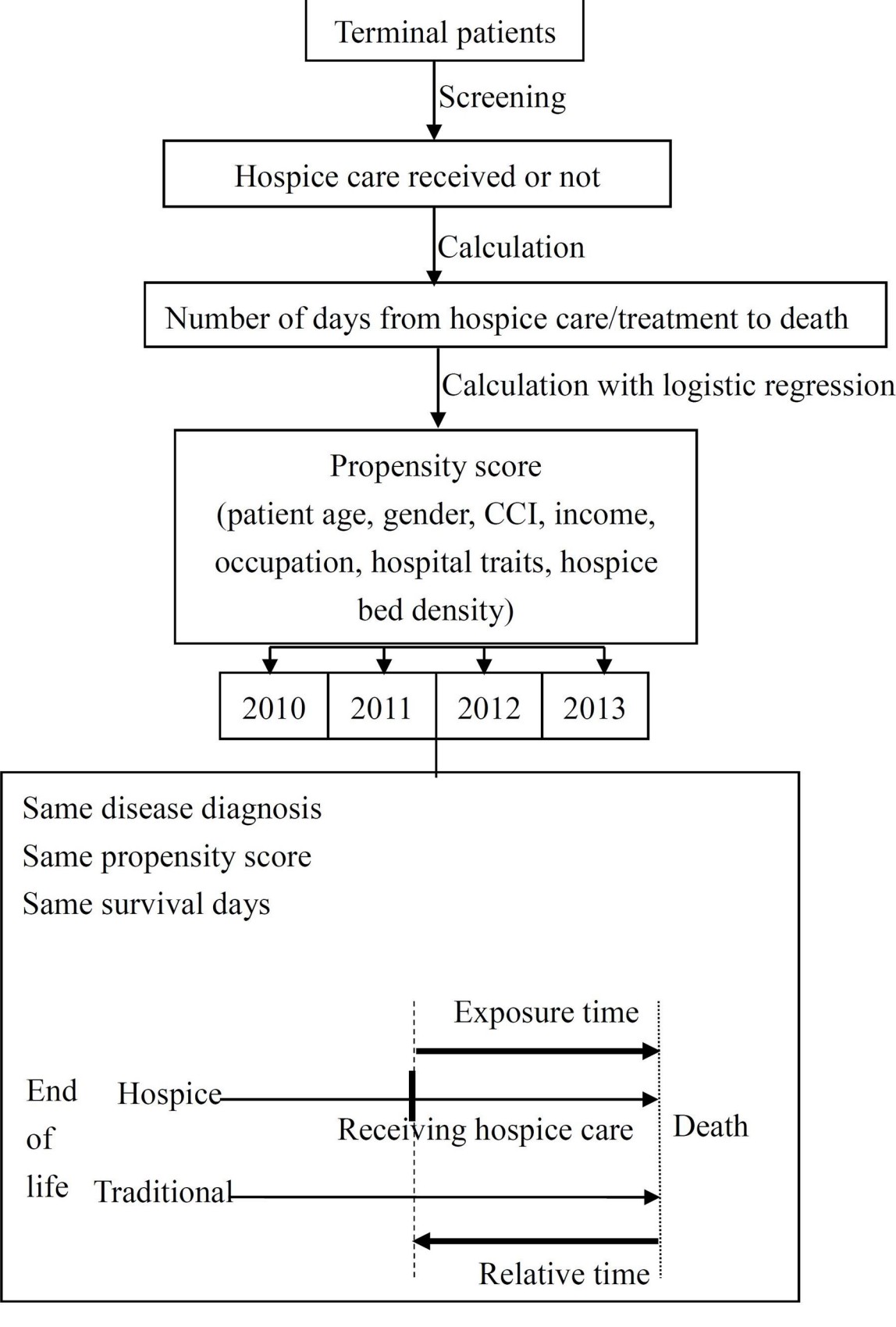

**Fig 1. Trends in the proportion of terminal patients receiving hospice care.**

the greatest number of hospitalizations in the year before the death of the terminal patients. If there were no hospitalization data, this would be the hospital with the most number of outpatient visits, its area, hospital level, number of hospital beds, and number of hospice beds [7, 39]. The value of hospice bed density indicated the availability of hospice services. Density was mainly used to calculate the proportion of hospice beds to the total number of cancer deaths per year in each hospital [7, 39]. Medical treatment data of the terminally ill patients before death included use of outpatient and emergency services; inpatient medical use from treatment/hospice to death, including number of days spent in an acute bed, whether they had received surgical treatment, chemotherapy, respirator treatment, hemodialysis treatment, or cardiopulmonary resuscitation (CPR), admission into the intensive care unit (ICU), and the place of death.

The data of pre-death medical expenses were used to calculate the medical expenses from receiving treatment/hospice to death, including the total medical expenses, the total medical expenses for outpatient (emergency) care, and the total medical expenses for hospitalization. This study was reviewed and approved by the Institutional Review Board of Camillian Saint Mary's Hospital Luodong (approval # SMHIRB104004). The study design conformed to the ethical guidelines of the 1975 Declaration of Helsinki. The written informed consent was waived because the database was provided and analyzed anonymously.

## Data analysis

In the analysis of the data, SPSS 22.0 (IBM, Armonk, NY, USA) and R (R Foundation for Statistical Computing, Vienna, Austria) software were used. Basic demographic information, hospital characteristics, the density of hospice beds, pre-death medical treatment behavior, and medical expenses included the average, standard deviation, and percentage. The Cochran-Armitage trend test was used to verify the usage trend of hospice care from 2010 to 2013. Then the differences between the traditional group and the hospice group were analyzed using a generalized linear mixed-effects model. In this model, the random clustering effect of hospitals and the year of death were used to control the habitual influence of treatment given by each hospital in each year. If the data was a category variable (e.g., the proportion of patients with medical-seeking behavior), the adjusted odds ratio (AOR) was calculated. When the AOR was less than 1, the odds ratio of hospice patients receiving treatment was lower than that of traditional patients; if it was greater than 1, the odds ratio of hospice patients receiving treatment was higher than that of traditional patients. Because medical expenses are continuous variables and present a gamma distribution, the log link function analysis of the generalized linear mixed-effects model and gamma distribution were performed, and the relative ratio (RR) and 95% confidence intervals were calculated. Subsequently, the average values for the traditional care and hospice groups, which used the random clustering effect of the hospital and the year of death to control the habitual influence of treatment given by each hospital in each year, were calculated using the least-squares means. Differences obtained by subtracting average values for the hospice group from those of the traditional group were the adjusted mean differences. When the adjusted mean difference was negative, it represented the medical expenses saved by the hospice group.

Finally, the cumulative hospice care savings were calculated using the total medical expenses of the terminal patients before and after receiving hospice care 1 year before death (i.e., the date of first receiving hospice care and the date of receiving treatment of their paired

traditional group subtracted from the date of hospitalization/visit), after which the following were calculated: (a) the number of days before hospice from the beginning of the terminal diagnosis to the date of receiving hospice (showing negative values), and (b) after hospice care, which referred to the number of days from receiving hospice care to death (positive value). Then the average daily expenses of hospice care and traditional care before and after hospice care were calculated, and the savings after receiving hospice care (hospice care medical expenses minus traditional care medical expenses) were calculated. Total accumulated expenses in the last year, regardless of the duration of hospice care, were calculated and defined as the cumulative hospice care savings (for example, the terminal patients began to receive hospice care 30 days before death, while the date of death for the patients in the traditional group was backdated by 30 days for matching, and the medical expenses of the two groups of patients started to accumulate from 335 days before receiving hospice care/treatment to death). Finally, the hospice care savings before and after hospice care were calculated in an accumulative manner 365 days before death to determine the cumulative cost saved, and the data were plotted on a graph.

## Results

This study retrospectively reviewed death cases from January 1, 2010, to December 31, 2013, and pushed back the date of death for one year. The total number of terminal cancer patients who died from 2010 to 2013 was 7,396, and 32.37% ($n$ = 2,394) of these patients received hospice care more than once during the year before death. Over four years, the proportion of the population receiving hospice care (hospice group) increased from 22.2% to 41.30% ($p < .001$) (Table 1 and Fig 2). However, 67.63% ($n$ = 5,002) never received any form of hospice service (traditional group). After matching, both the hospice group and the traditional group consisted of 1,774 terminal cancer patients (Fig 3).

### Basic characteristics of terminal stage patients in traditional care or hospice care

The average age of the patients in the hospice group was 68.97±13.71 years, and the percentage of male patients was 60.65%. The average CCI was 10.75±4.43. The average age of the patients

**Table 1. Proportions of the different types of hospice care provided to terminal patients[a].**

| [$n$ = person (%)] | 2010 | 2011 | 2012 | 2013 | $p$ for Trend Test | Total |
|---|---|---|---|---|---|---|
| Hospice care[b] | 371 (22.2) | 499 (27.15) | 719 (37.06) | 805 (41.35) | < .001 | 2,394 (32.37) |
| Inpatient hospice care[b] | 309 (18.49) | 336 (18.28) | 363 (18.71) | 355 (18.23) | .931 | 1,363 (18.43) |
| Home-based hospice care[b] | 151 (9.04) | 146 (7.94) | 188 (9.69) | 195 (10.02) | .108 | 680 (9.19) |
| Hospice-shared care[b] | 0 (0) | 167 (9.09) | 469 (24.18) | 579 (29.74) | < .001 | 1,215 (16.43) |
| **Hospice type[b]** | | | | | | |
| No hospice | 1,300 (77.8) | 1,339 (72.9) | 1,221 (62.90) | 1,142 (58.7) | | 5,002 (67.60) |
| Hospice-shared care | 0 (0) | 99 (5.4) | 278 (14.3) | 373 (19.2) | < .001 | 750 (10.10) |
| Inpatient hospice care | 220 (13.2) | 212 (11.5) | 133 (6.9) | 117 (6) | | 682 (9.20) |
| Home-based hospice car | 62 (3.7) | 52 (2.8) | 47 (2.4) | 39 (2) | | 200 (2.70) |
| Hospice-shared care and inpatient hospice care | 0 (0) | 42 (2.3) | 120 (6.2) | 120 (6.2) | | 282 (3.8) |
| Hospice-shared care and home-based hospice care | 0 (0) | 12 (0.7) | 31 (1.6) | 38 (2) | | 81 (1.1) |
| Inpatient and home-based hospice care | 89 (5.3) | 68 (3.7) | 70 (3.6) | 70 (3.6) | | 297 (4) |
| Combination of the above three | 0 (0) | 14 (0.8) | 40 (2.1) | 48 (2.5) | | 102 (1.4) |

[a] The usage trend of hospice care from 2010 to 2013.

[b] Differences in the proportions for the years are the inferential statistics based on the Cochran-Armitage trend test.

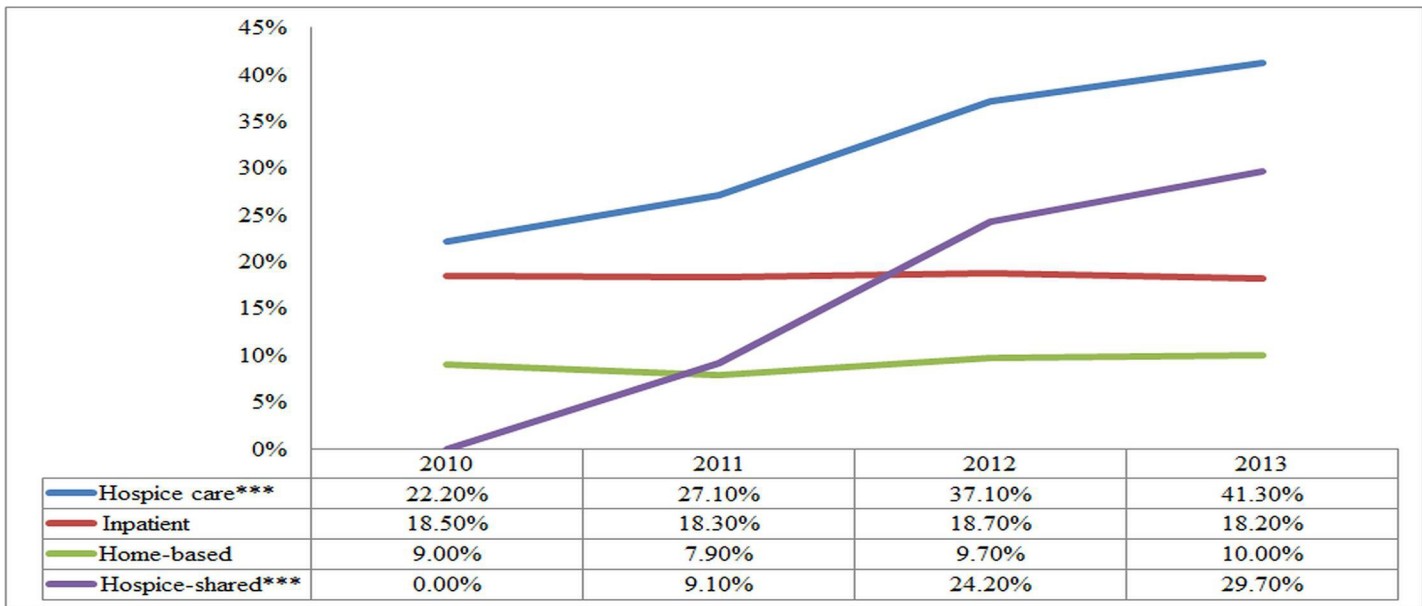

| | 2010 | 2011 | 2012 | 2013 |
|---|---|---|---|---|
| Hospice care*** | 22.20% | 27.10% | 37.10% | 41.30% |
| Inpatient | 18.50% | 18.30% | 18.70% | 18.20% |
| Home-based | 9.00% | 7.90% | 9.70% | 10.00% |
| Hospice-shared*** | 0.00% | 9.10% | 24.20% | 29.70% |

**Fig 2. Flow chart of the sampling and matching process.** CCI = Charlson Comorbidity Index.

in the traditional group was 68.69±13.48 years, and the percentage of male patients was 60.43%. The average CCI was 10.65±4.78. There was no significant difference between the two groups ($p > .05$). Of all the patients in the hospice group, 45.56% had an average monthly income ranging from US$300 to US$1,000, which was lower than that in the traditional group. However, there was no significant difference ($p = .049$) (Table 2).

### Medical treatment behaviors of terminally ill patients seeking traditional care or hospice care

During the period from hospice care to death, the rates of outpatient and emergency treatment, ICU use, ventilator use, CPR administration, hemodialysis, surgery, and chemotherapy use were all significantly lower than those in the traditional group ($p < .05$). However, during the period from receiving hospice care to death, 55.30% (AOR = 2.00) of the hospice group patients were hospitalized for more than 14 days, and the proportion of hospital deaths was 73.00% (AOR = 1.43). These percentages were both significantly higher than those of the traditional group ($p < .05$) (Table 3).

### Medical expenses for terminal patients in traditional care or hospice care

One year before death, the average total medical expenses in the hospice group (using an exchange rate of 1 US dollar = 31 Taiwan dollars) was US$17,821.33±12,779.1 per person, whereas, in the traditional group, the average total medical expenses was US$17,558.22 ±12,133.85 per person (RR 1.02, $p = .492$). In the month before receiving hospice care, the average total medical expenses in the hospice group were US$2,439.06±2,722.13 per person, whereas, in the traditional group, the average total medical expenses were US$1,456.14 ±1,996.25 per person (RR 1.68, $p < .001$). However, from the time of receiving hospice care to the time of death, the average total medical expense per person in the hospice group was US $3,784.76±5,487.59, whereas the average total medical expenses per person in the traditional group was US$5,236.59±7,082.12. After adjustment, US$1,455.30 (RR 0.71, 95% CI 0.66~0.76)

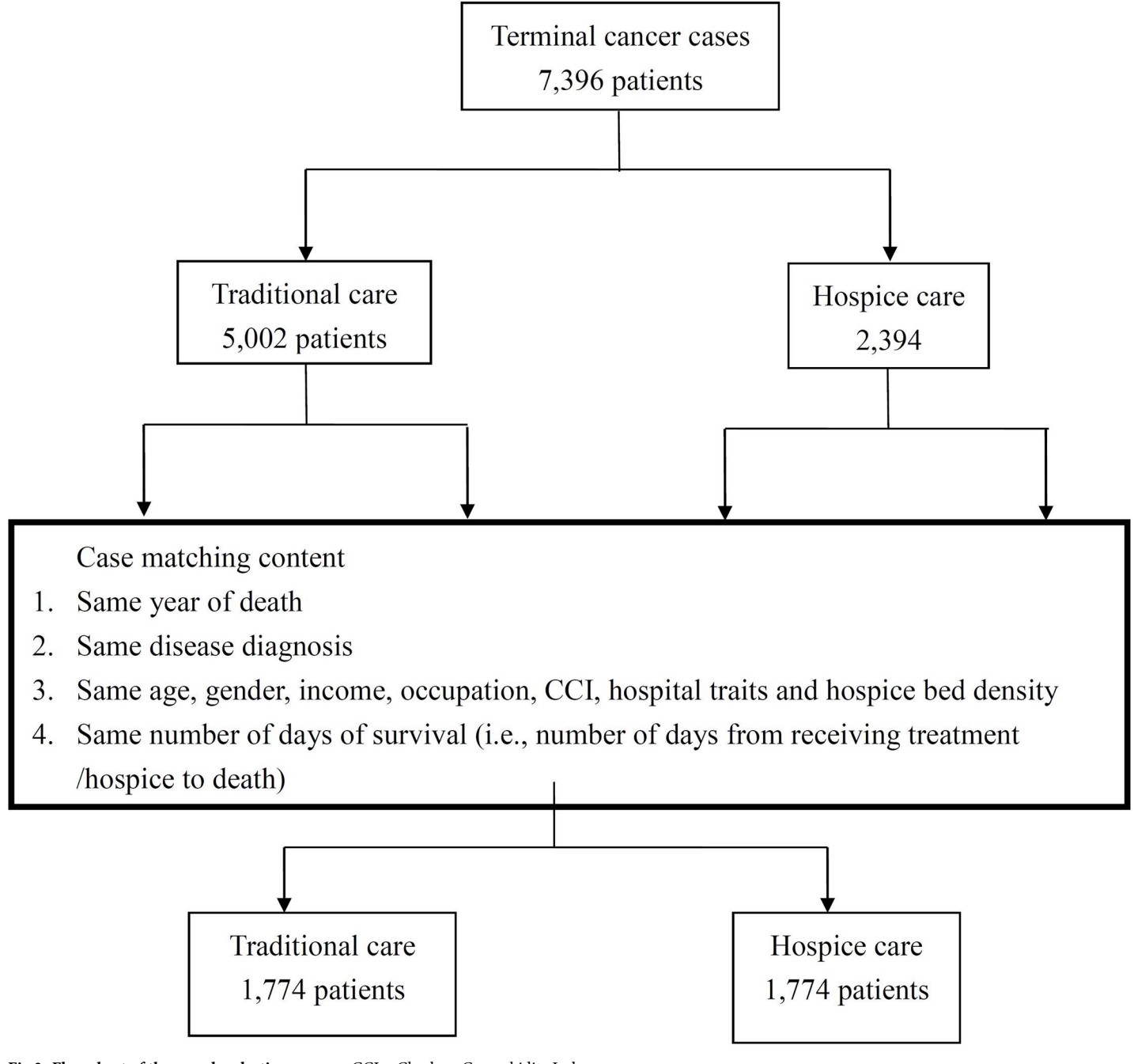

**Fig 3. Flow chart of the sample selection process.** CCI = Charlson Comorbidity Index.

of the total medical expenses could be saved for each person in hospice care (Table 4 and Fig 4). In addition, the total medical expenses in a different kind of hospice care could also be saved. However, the savings receiving hospice-shared care and combination of the three hospice care were insignificant (Table 5 and Fig 5).

The concept of cumulative cost savings was further used to analyze the cumulative cost-saving trend of total medical expenses before and after receiving hospice care. It was found that cumulative medical expenses for the patients in the hospice group reached their peak (US

**Table 2. Basic characteristics of terminal patients receiving traditional care or hospice care[a].**

| | | 1–3 days | 4–7 days | 8–14 days | 15–30 days | 31–60 days | 61–90 days | 91–180 days | 181–365 days | All |
|---|---|---|---|---|---|---|---|---|---|---|
| Number of cases | Traditional | 122 | 141 | 369 | 407 | 365 | 148 | 137 | 85 | 1,774 |
| | Hospice | 122 | 141 | 369 | 407 | 365 | 148 | 137 | 85 | 1,774 |
| Gender[b] (Male) | Traditional | 76 (62.30) | 88 (62.41) | 233 (63.14) | 244 (59.95) | 216 (59.18) | 85 (57.43) | 84 (61.31) | 46 (54.12) | 1,072 (60.43) |
| | Hospice | 76 (62.30) | 87 (61.70) | 243 (65.85) | 236 (57.99) | 215 (58.9) | 93 (62.84) | 80 (58.39) | 46 (54.12) | 1,076 (60.65) |
| | p | 1.000 | .902 | .442 | .569 | .940 | .342 | .622 | 1.000 | .891 |
| Age[c] | Traditional | 68.75±13.57 | 69.33±14.1 | 68.50±13.21 | 68.84±13.37 | 67.83±13.44 | 68.94±12.76 | 68.14±14.78 | 71.81±13.17 | 68.69±13.48 |
| | Hospice | 68.65±13.48 | 70.04±13.2 | 67.92±12.78 | 69.26±14.18 | 68.46±13.93 | 70.17±13.48 | 68.15±15.14 | 72.34±13.23 | 68.97±13.71 |
| | p | .984 | .591 | .974 | .788 | .628 | .544 | .895 | .813 | .532 |
| CCI[c] | Traditional | 10.92±5.14 | 10.30±4.6 | 10.01±4.59 | 10.57±4.7 | 11.20±4.64 | 10.95±4.66 | 11.00±5.03 | 10.51±5.74 | 10.65±4.78 |
| | Hospice | 11.60±4.65 | 10.57±4.24 | 10.28±4.18 | 10.66±4.48 | 11.21±4.12 | 10.90±4.93 | 10.28±4.74 | 10.94±4.78 | 10.75±4.43 |
| | p | .286 | .254 | .152 | .281 | .568 | .839 | .185 | .497 | .059 |
| Income[b] (US$ 300–1000) | Traditional | 34 (50.00) | 42 (49.41) | 121 (49.79) | 137 (55.02) | 133 (54.96) | 49 (51.04) | 30 (34.48) | 18 (40.00) | 564 (50.58) |
| | Hospice | 37 (50.68) | 47 (54.65) | 107 (44.77) | 123 (49.2) | 92 (42.59) | 38 (40.86) | 32 (41.03) | 17 (36.17) | 493 (45.56) |
| | p | .990 | .585 | .537 | .184 | .030 | .266 | .197 | .608 | .049 |
| Hospital level[b] (medical center) | Traditional | 65 (53.28) | 76 (53.90) | 208 (56.37) | 226 (55.53) | 186 (50.96) | 80 (54.05) | 64 (46.72) | 32 (37.65) | 937 (52.82) |
| | Hospice | 65 (53.28) | 76 (53.90) | 208 (56.37) | 225 (55.28) | 187 (51.23) | 80 (54.05) | 64 (46.72) | 32 (37.65) | 937 (52.82) |
| | p | 1.000 | 1.000 | 1.000 | .944 | .941 | 1.000 | 1.000 | 1.000 | 1.000 |
| Hospital area[b] (North) | Traditional | 46 (37.70) | 60 (42.55) | 150 (40.65) | 161 (39.56) | 146 (40.00) | 61 (41.22) | 58 (42.34) | 35 (41.18) | 717 (40.42) |
| | Hospice | 63 (51.64) | 41 (29.08) | 162 (43.90) | 172 (42.26) | 146 (40.00) | 54 (36.49) | 58 (42.34) | 34 (40.00) | 730 (41.15) |
| | p | .029 | .018 | .371 | .433 | 1.000 | .404 | 1.000 | .876 | .657 |
| Hospice bed density[c] | Traditional | 0.42±0.96 | 0.38±0.8 | 0.3±0.5 | 0.51±1.33 | 0.46±0.96 | 0.43±0.88 | 0.38±0.81 | 0.57±1.15 | 0.43±0.97 |
| | Hospice | 0.33±0.49 | 0.44±0.71 | 0.33±0.51 | 0.48±0.77 | 0.52±0.87 | 0.44±0.68 | 0.49±0.74 | 0.57±1.36 | 0.45±0.76 |
| | p | .690 | .514 | .821 | .552 | .571 | .369 | .725 | .793 | .893 |

CCI = Charlson Comorbidity Index.

[a]The number of days of survival was equal to the days from the date of receiving hospice care or treatment to the date of death.

[b]The differences between the two groups for the category variables was determined by the chi-square test.

[c]If the continuous variable was the abnormal distribution, the log conversion of the variable was carried out, and the difference between the two groups was analyzed by independent t test.

$1,295 per person) on the day of receiving hospice care (Fig 6). As the number of days of survival after receiving hospice care increased, cumulative medical expenses showed a gradual downward trend until the patients had survived for more than 61 days. A comparison of cumulative medical expenses for the traditional group with those of the hospice group demonstrated that the hospice group was beginning to save money, with an average of US$200 saved for one person and a maximum of US$6,497 saved for each person who survived for 365 days (Fig 6).

## Discussion

In recent years, hospice care has accounted for 9% to 48% of the total number of service users around the world in countries such as the United States, the United Kingdom, Canada, Australia, Japan, and Korea [8, 41–49]. Previously, although the utilization rate of hospice care in Taiwan was slightly higher in comparison to other Asian countries such as Japan [46] and Korea (at 9%–12%) [43, 47], it is much lower compared with countries in Europe and North America, such as the United States (at 48%) [50], the United Kingdom (at 44%) [51], and Canada (at 30%) [41]. The present study and that of Shao et al. [7] both confirmed that the utilization rate

**Table 3. Differences in medical treatment behaviors between traditional care and hospice care patients[a].**

| | | 1–3 days | 4–7 days | 8–14 days | 15–30 days | 31–60 days | 61–90 days | 91–180 days | 181–365 days | All |
|---|---|---|---|---|---|---|---|---|---|---|
| Number of cases | Traditional | 122 | 141 | 369 | 407 | 365 | 148 | 137 | 85 | 1,774 |
| | Hospice | 122 | 141 | 369 | 407 | 365 | 148 | 137 | 85 | 1,774 |
| Emergency visits[a] | Traditional | 67 (54.92) | 75 (53.19) | 203 (55.01) | 238 (58.48) | 229 (62.74) | 96 (64.86) | 99 (72.26) | 64 (75.29) | 1,071 (60.37) |
| | Hospice | 33 (27.05) | 49 (34.75) | 101 (27.37) | 140 (34.40) | 170 (46.58) | 91 (61.49) | 84 (61.31) | 56 (65.88) | 724 (40.81) |
| | AOR | 0.27 (0.15–0.50) *** | 0.44 (0.25–0.77) ** | 0.25 (0.18–0.36) *** | 0.32 (0.24–0.45) *** | 0.43 (0.31–0.61) *** | 0.76 (0.43–1.34) | 0.57 (0.57–0.57) *** | 0.47 (0.19–1.13) | 0.38 (0.33–0.44) *** |
| Outpatient visits[a] | Traditional | 47 (38.52) | 91 (64.54) | 249 (67.48) | 316 (77.64) | 339 (92.88) | 145 (97.97) | 135 (98.54) | 85 (100) | 1,407 (79.31) |
| | Hospice | 33 (27.05) | 36 (25.53) | 112 (30.35) | 197 (48.40) | 241 (66.03) | 134 (90.54) | 130 (94.89) | 85 (100) | 968 (54.57) |
| | AOR | 0.59 (0.34–1.02) | 0.18 (0.1–0.33) *** | 0.20 (0.15–0.28) *** | 0.25 (0.18–0.35) *** | 0.13 (0.13–0.13) *** | 0.17 (0.04–0.66) * | 0.27 (0.05–1.41) | – | 0.29 (0.25–0.34) *** |
| Inpatient treatment[a] | Traditional | 75 (61.48) | 98 (69.50) | 281 (76.15) | 354 (86.98) | 337 (92.33) | 140 (94.59) | 132 (96.35) | 83 (97.65) | 1,500 (84.55) |
| | Hospice | 108 (88.52) | 121 (85.82) | 242 (65.58) | 359 (88.21) | 336 (92.05) | 143 (96.62) | 131 (95.62) | 82 (96.47) | 1,522 (85.79) |
| | AOR | 4.85 (2.49–9.45) | 2.74 (1.48–5.08) ** | 0.57 (0.41–0.79) *** | 1.11 (0.73–1.70) | 0.97 (0.56–1.68) | 1.67 (0.51–5.44) | 0.41 (0.04–3.94) | 5.97 (0.01–8.66) | 1.09 (0.9–1.32) |
| ICU treatment[a] | Traditional | 29 (23.77) | 34 (24.11) | 96 (26.02) | 101 (24.82) | 115 (31.51) | 46 (31.08) | 54 (39.42) | 42 (49.41) | 517 (29.14) |
| | Hospice | 4 (3.28) | 11 (7.80) | 8 (2.17) | 21 (5.16) | 32 (8.77) | 18 (12.16) | 22 (16.06) | 13 (15.29) | 129 (7.27) |
| | AOR | 0.10 (0.03–0.31) *** | 0.27 (0.27–0.28) *** | 0.06 (0.03–0.12) *** | 0.16 (0.1–0.27) *** | 0.20 (0.13–0.31) *** | 0.30 (0.16–0.56) *** | 0.29 (0.17–0.52) *** | 0.18 (0.08–0.39) *** | 0.18 (0.15–0.22) *** |
| Respirator treatment[a] | Traditional | 24 (19.67) | 24 (17.02) | 78 (21.14) | 91 (22.36) | 94 (25.75) | 37 (25.00) | 44 (32.12) | 33 (38.82) | 425 (23.96) |
| | Hospice | 2 (1.64) | 4 (2.84) | 4 (1.08) | 11 (2.70) | 14 (3.84) | 9 (6.08) | 9 (6.57) | 5 (5.88) | 58 (3.27) |
| | AOR | 0.06 (0.01–0.28) *** | 0.14 (0.05–0.42) *** | 0.04 (0.01–0.11) *** | 0.10 (0.05–0.18) *** | 0.11 (0.06–0.20) *** | 0.20 (0.09–0.43) *** | 0.14 (0.07–0.32) *** | 0.10 (0.04–0.27) *** | 0.11 (0.08–0.14) *** |
| CPR[a] | Traditional | 21 (17.21) | 13 (9.22) | 47 (12.74) | 41 (10.07) | 38 (10.41) | 16 (10.81) | 11 (8.03) | 3 (3.53) | 190 (10.71) |
| | Hospice | 3 (2.46) | 3 (2.13) | 2 (0.54) | 4 (0.98) | 6 (1.64) | 2 (1.35) | 2 (1.46) | 1 (1.18) | 23 (1.30) |
| | AOR | 0.11 (0.03–0.4) *** | 0.07 (0.02–0.28) *** | 0.04 (0.01–0.15) *** | 0.09 (0.03–0.25) *** | 0.14 (0.06–0.33) *** | 0.09 (0.02–0.47) ** | 0.17 (0.04–0.78) * | 0.35 (0.01–18.33) | 0.11 (0.07–0.17) *** |

(*Continued*)

**Table 3.** (Continued)

|  |  | 1–3 days | 4–7 days | 8–14 days | 15–30 days | 31–60 days | 61–90 days | 91–180 days | 181–365 days | All |
|---|---|---|---|---|---|---|---|---|---|---|
| Hemodialysis[a] | Traditional | 4 (3.28) | 13 (9.22) | 31 (8.40) | 34 (8.35) | 31 (8.49) | 11 (7.43) | 15 (10.95) | 14 (16.47) | 153 (8.62) |
|  | Hospice | 1 (0.82) | 4 (2.84) | 2 (0.54) | 6 (1.47) | 7 (1.92) | 4 (2.7) | 1 (0.73) | 1 (1.18) | 26 (1.47) |
|  | AOR | 0.38 (0.02–5.8) | 0.40 (0.15–1.04) | 0.06 (0.01–0.25) *** | 0.16 (0.07–0.39) *** | 0.21 (0.09–0.48) *** | 0.33 (0.1–1.13) | 0.03 (0–0.38) ** | 0.04 (0–0.43) ** | 0.16 (0.1–0.24) *** |
| Chemotherapy[a] | Traditional | 2 (1.64) | 7 (4.96) | 24 (6.50) | 65 (15.97) | 84 (23.01) | 37 (25.00) | 45 32.85) | 31 (36.47) | 295 (16.63) |
|  | Hospice | 0 (0) | 1 (0.71) | 3 (0.81) | 10 (2.46) | 23 (6.30) | 13 (8.78) | 17 (12.41) | 12 (14.12) | 79 (4.45) |
|  | AOR | 0.05 (0.01–0.37) * | 0.06 (0–0.83) * | 0.11 (0.03–0.38) *** | 0.13 (0.06–0.25) *** | 0.21 (0.13–0.35) *** | 0.28 (0.14–0.57) *** | 0.25 (0.12–0.51) *** | 0.27 (0.11–0.62) ** | 0.23 (0.18–0.3) *** |
| Surgery[a] | Traditional | 19 (15.57) | 43 (30.50) | 122 (33.06) | 183 (44.96) | 212 (58.08) | 103 (69.59) | 98 (71.53) | 67 (78.82) | 847 (47.75) |
|  | Hospice | 5 (4.10) | 22 (15.60) | 38 (10.30) | 118 (28.99) | 144 (39.45) | 60 (40.54) | 64 (46.72) | 48 (56.47) | 499 (28.13) |
|  | AOR | 0.23 (0.08–0.67) ** | 0.39 (0.21–0.72) ** | 0.22 (0.15–0.33) *** | 0.49 (0.36–0.66) *** | 0.47 (0.35–0.64) *** | 0.28 (0.17–0.47) *** | 0.31 (0.18–0.54) *** | 0.31 (0.15–0.67) ** | 0.41 (0.36–0.48) *** |
| Hospitalization for more than 14 days[a] | Traditional | 0 (0) | 0 (0) | 15 (4.07) | 166 (40.79) | 222 (60.82) | 103 (69.59) | 105 (76.64) | 71 (83.53) | 682 (38.44) |
|  | Hospice | 0 (0) | 0 (0) | 114 (30.89) | 266 (65.36) | 281 (76.99) | 124 (83.78) | 118 (86.13) | 70 (82.35) | 981 (55.30) |
|  | AOR | – | – | 10.98 (6.22–9.40) *** | 2.8 (2.10–3.73) *** | 2.2 (1.58–3.06) *** | 2.31 (1.28–4.15) ** | 1.92 (1.01–3.64) * | 1.04 (1.03–1.05) *** | 1.95 (1.7–2.23) *** |
| Death in hospital[a] | Traditional | 68 (55.74) | 88 (62.41) | 225 (60.98) | 273 (67.08) | 241 (66.03) | 102 (68.92) | 103 (75.18) | 61 (71.76) | 1,161 (65.45) |
|  | Hospice | 93 (76.23) | 112 (79.43) | 300 (81.3) | 277 (68.06) | 263 (72.05) | 106 (71.62) | 92 (67.15) | 52 (61.18) | 1,295 (73.00) |
|  | AOR | 2.69 (1.51–4.79) *** | 2.57 (1.43–4.63) ** | 2.78 (1.97–3.91) *** | 1.05 (0.77–1.42) | 1.32 (0.96–1.83) | 1.14 (0.69–1.88) | 0.66 (0.38–1.13) | 0.62 (0.31–1.22) | 1.43 (1.24–1.66) *** |

AOR = adjusted odds ratio, CPR = cardiopulmonary resuscitation, ICU = intensive care unit.

[a]The difference analysis of the two groups was performed using the generalized linear mixed-effects model. In this model, the death year and hospital attributes were placed in the random effects model for control. The AOR correction is the calculated odds ratio after controlling the habitual effect of treatment in each hospital in each year. If the AOR is less than 1, the odds ratio for the patients in hospice care to receive treatment is lower than that of the patients of traditional care; if the AOR is greater than 1, the odds ratio for the patients in hospice care to receive treatment is higher than that of the patients of traditional care.

* $p < .05$

** $p < .01$

*** $p < .001$

of hospice care rose rapidly from 27%–28% to 41%–42% (rates similar to those of European and North American countries) after the implementation hospice-shared care of the program.

**Table 4. Differences in medical expenses between traditional care and hospice care patients[a].**

| | | 1–3 days | 4–7 days | 8–14 days | 15–30 days | 31–60 days | 61–90 days | 91–180 days | 181–365 days | All |
|---|---|---|---|---|---|---|---|---|---|---|
| Number of cases | Traditional | 122 | 141 | 369 | 407 | 365 | 148 | 137 | 85 | 1,774 |
| | Hospice | 122 | 141 | 369 | 407 | 365 | 148 | 137 | 85 | 1,774 |
| Total medical expenses[a] | Traditional | 1,099.32± 2,791.49 | 1,947.6± 4,715.85 | 2,269.61± 2,834.53 | 3,544.74± 3,513.23 | 5,916.77± 5,489.81 | 8,618.16± 7,501.79 | 12,042.91± 9,657.75 | 17,833± 13,475.44 | 5,236.59± 7,082.12 |
| | Hospice | 434.28± 530.59 | 1,036.63± 821.83 | 1,920.12± 2,008.9 | 2,360.33± 4,632.57 | 4,091.83± 3,389.94 | 6,097.71± 4,644.47 | 8,972.85± 7,473.26 | 14,359.87± 10,663.96 | 3,784.76± 5,487.59 |
| | AMD | -477.76 | -452.6 | -344.49 | -1,287.33 | -1,762.26 | -2,078.97 | -2,645.76 | -900.87 | -1,455.30 |
| | RR(CI) | 0.44 (0.44~0.44) *** | 0.70 (0.55~0.89) ** | 0.84 (0.73~0.95) ** | 0.63 (0.55~0.72) *** | 0.69 (0.68~0.71) *** | 0.74 (0.61~0.89) ** | 0.74 (0.61~0.9) ** | 0.71 (0.66~0.76) *** | 0.71 (0.66~0.76) *** |

AMD = adjusted mean difference, CI = confidence interval, RR = relative ratio.

[a]The distribution of medical expenses presented a gamma distribution. Therefore, the differential verification of the two groups was performed with the generalized linear mixed-effect model and the log-link function of the gamma distribution. In this model, the year of death and hospital attributes are placed in the random effects model for control. After the average values for the traditional care and hospice groups are calculated by least squares means in controlling the treatment of each hospital in each year, the differences between the average values for traditional and hospice groups are the AMDs. A negative AMD indicates savings of medical expenses. One US dollar = thirty-one Taiwan dollars.

* $p < .05$

** $p < .01$

*** $p < .001$

In past studies, the proportions of medical utilization such as outpatient treatment, emergency treatment, intensive care units, respirators, CPR, hemodialysis, surgery, and chemotherapy were all reduced due to the use of hospice care services [7, 8, 30, 52–62]. These findings are consistent with those of the present study. In the present study, the hospitalization rate,

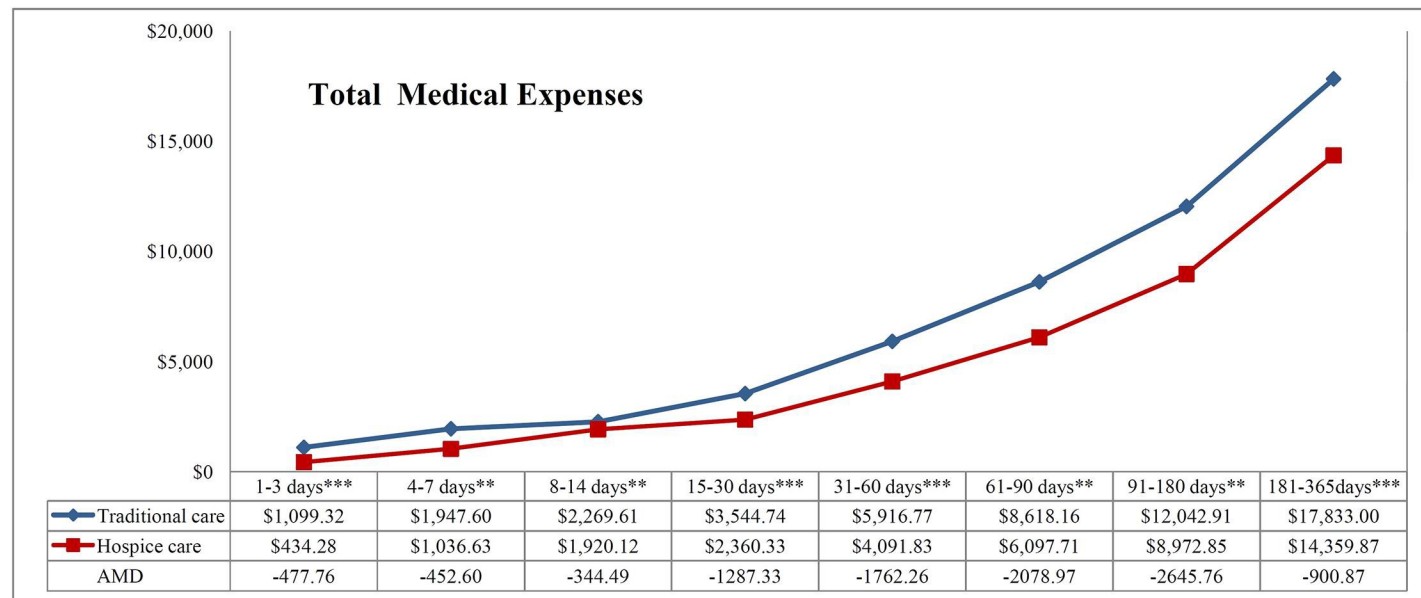

**Fig 4. Total medical expenses for traditional care and hospice care.** After the average values for the traditional group and the hospice group were calculated by the least-squares means in controlling the treatment of each hospital in each year, the differences between the average values of the hospice treatment and the traditional group were the adjusted mean difference (AMDs). Negative AMD values indicate savings on medical expenses. 1 US dollar = 31 Taiwan Dollars; * $p < .05$; ** $p < .01$; *** $p < .001$.

**Table 5. Differences in medical expenses between traditional care and various hospice care patients[a].**

|  |  | HSC | IHC | HHC | HSC+IHC | HSC+HHC | IHC+HHC | HSC+IHC+HHC |
|---|---|---|---|---|---|---|---|---|
| Number of cases | Traditional | 523 | 508 | 164 | 198 | 63 | 236 | 82 |
|  | Hospice | 523 | 508 | 164 | 198 | 63 | 236 | 82 |
| Total medical expenses[a] | Traditional | 3920.04± 5769.32 | 3782.94± 5374.91 | 6886.29± 10132.45 | 5116.39± 5699.17 | 6465.09± 7376.92 | 8497.94± 8639.27 | 9299.87± 8937.45 |
|  | Hospice | 3749.22± 6653.09 | 3379.72± 4477.74 | 3008.04± 5414.4 | 3207.69± 4076.28 | 3150.43± 2932.57 | 4683.61± 5074.39 | 7368.02± 6997.8 |
|  | AMD | -201.85 | -433.65 | -4013.92 | -1891.33 | -2939.05 | -3476.9 | -1523.18 |
|  | RR(CI) | 0.95 (0.83~1.08) | 0.88 (0.78~1.01) * | 0.35 (0.35~0.36) *** | 0.63 (0.51~0.78) *** | 0.49 (0.35~0.71) *** | 0.54 (0.45~0.64) *** | 0.81 (0.61~1.08) |

AMD = adjusted mean difference, CI = confidence interval, HSC = Hospice-shared care; HHC = Home-based hospice care, IHC = Inpatient hospice care. RR = relative ratio

[a]The distribution of medical expenses presented a gamma distribution. Therefore, the differential verification of the two groups was performed with the generalized linear mixed-effect model and the log-link function of the gamma distribution. In this model, the year of death and hospital attributes are placed in the random effects model for control. After the average values for the traditional care and hospice groups are calculated by least squares means in controlling the treatment of each hospital in each year, the differences between the average values for traditional and hospice groups are the AMDs. A negative AMD indicates savings of medical expenses. One US dollar = thirty-one Taiwan dollars.

* $p < .05$

** $p < .01$

*** $p < .001$

hospitalization for more than 14 days, and the in-hospital death rate were higher in terminally ill patients receiving hospice care compared with those receiving traditional care. These findings were similar to those of Chang et al., Chiang et al., and Shao et al. [7, 30, 53, 55], but contradicted those of Obermeyer et al., Kelley et al., and Taylor et al. [32, 33, 63]. As the present

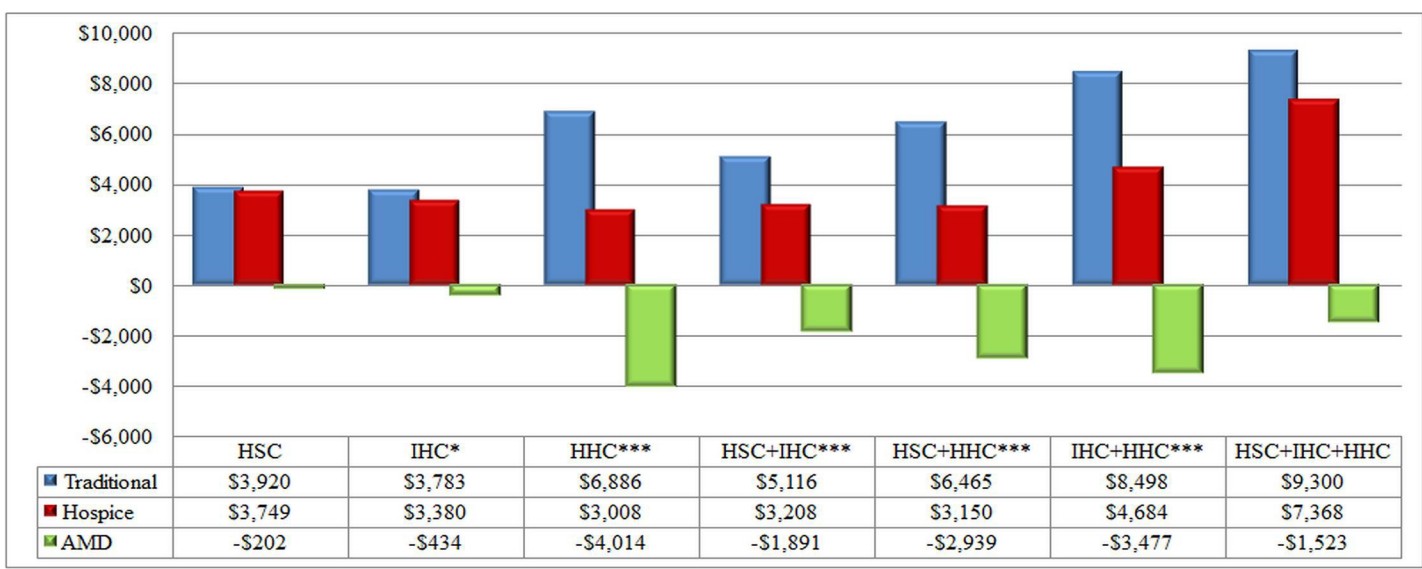

**Fig 5. Total medical expenses for traditional care and various hospice care.** After the average values for the traditional group and the hospice group were calculated by the least-squares means in controlling the treatment of each hospital in each year, the differences between the average values of the hospice treatment and the traditional group were the adjusted mean difference (AMDs). Negative AMD values indicate savings on medical expenses. 1 US dollar = 31 Taiwan Dollars; * $p < .05$; ** $p < .01$; *** $p < .001$. **Abbreviation:** HSC = Hospice-shared care; IHC = Inpatient hospice care; HHC = Home-based hospice care.

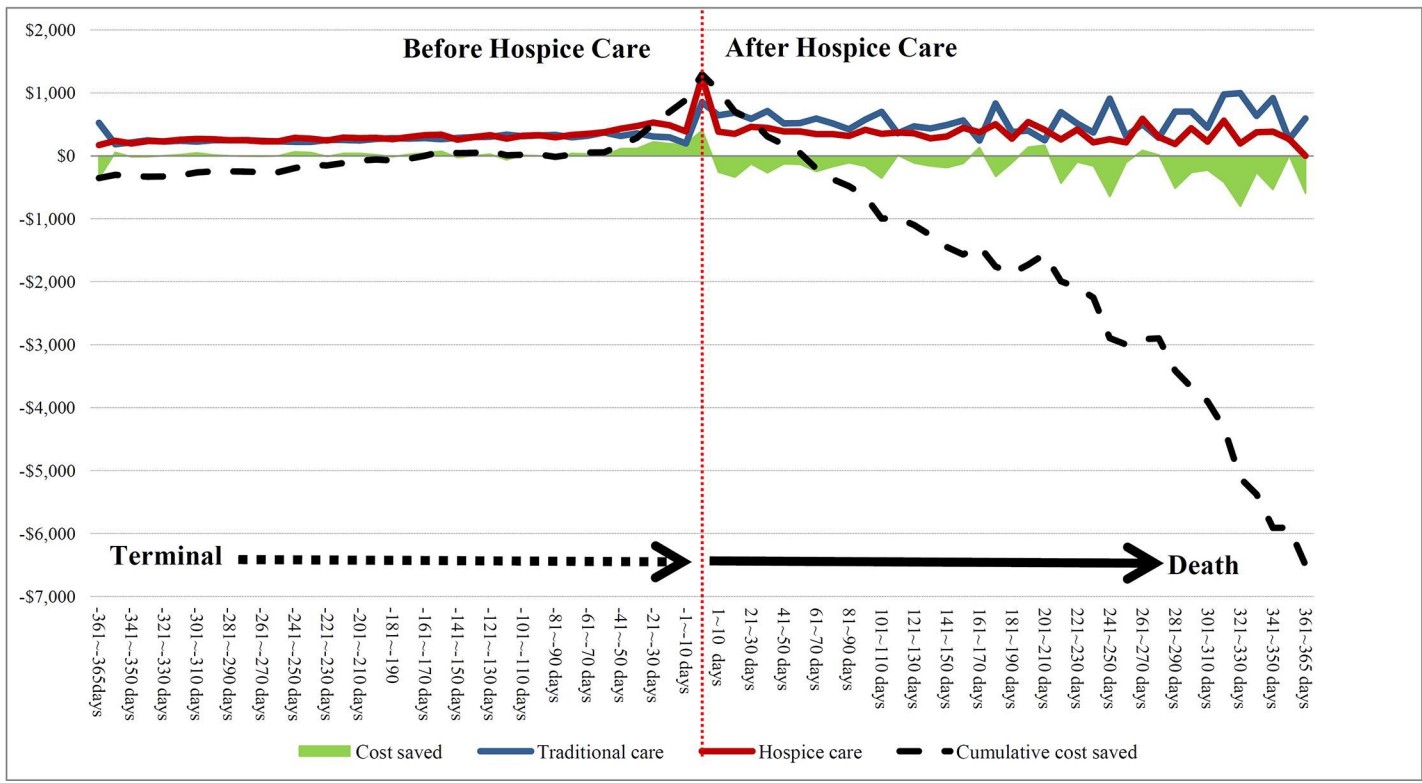

**Fig 6. Trends in medical costs before and after hospice care in the year before death.** The vertical dashed red line represents the day of receiving hospice care (i.e., point zero). The left side of the dashed red line represents the number of days from the beginning of the end of life to the day before accepting hospice care, and the right side of the dashed red line represents the number of days from accepting hospice care to death. The cumulative total cost for the last year was calculated regardless of the length of hospice care exposure time (e.g., terminal patients started receiving hospice care 30 days before death, while the date of death was pushed back 30 days in the traditional group for matching, and the medical cost for the two groups of patients started to accumulate to death from 335 days before receiving hospice care/ treatment). 1 US dollar = 31 Taiwan dollars.

study and the studies of Chang et al., Chiang et al., and Shao et al. all come from Taiwan, the findings were relatively similar. Taiwan's health care fees are low, and the NHI system provides an allowance to patients with cancer. The development of hospice care relied on large hospitals setting up exclusive hospice wards. These factors, along with poor effectiveness of home-based hospice care and various subjective factors involving family members of terminal cancer patients in Taiwan [7, 53, 64–70], cause frequent hospitalization, prolonged hospitalization days, and in-hospital death in terminally ill patients receiving hospice care.

Many studies have found that hospice care can effectively reduce the utilization rate and cost of medical care [30, 32, 33, 61]. Taylor et al. [33] and Obermeyer et al. [32] both found that hospice care can save on the cost of terminal medical expenses in the year before death (US$2,309 per person and US$8,697 per person, respectively). Shao et al. [7] found that hospice care can save about US$696 1 month before death, and Hung et al. [39] found the value to be US$3,075 for the same time point. The present study confirmed that end-stage patients could effectively save US$1,455 per person in end-stage medical expenses after receiving hospice care. Although the results of the present study were similar to those of previous studies, the timepoints of the end-stage medical expenses in previous studies are mostly fixed (e.g., comparisons of the differences in medical expenses in 1 month, or one year before death, whether in hospice care or not). However, end-stage medical expenses at a fixed time are easily affected by the time of referral to hospice care. For example, if the time of referral to hospice

care is seven days before death, expenses in the month before death will be affected by the active treatment during the 23 days before the start of hospice care. Therefore, it is possible to underestimate the savings brought by hospice care [63, 71]. The present study focused on the number of days from the start of hospice care to death, improving the problem of underestimating medical expense savings in hospice care.

The present study further explained the impact of hospice care on medical expenses one year before death using the concept of cumulative cost savings. It was found that cumulative medical expenses in the hospice group on the day of receiving hospice care (point zero) were higher than those in the traditional group when medical expenses reached their peak (US $1,295/person). However, with the increase of survival days after receiving hospice care, the cumulative medical expenses showed a decreasing trend. Cumulative cost savings began to be observed when patients survived for more than 61 days (US$200 per person on average and up to US$6,497 per person when the patients survived up to 365 days). A study by Obermeyer et al. also used cumulative cost savings to analyze the benefits of hospice care on medical costs. In that study, an analysis of the cumulative medical cost savings, accumulated one year before death, during the weeks before and after hospice care revealed that each patient receiving hospice care started to save medical costs within seven days after receiving hospice care. The highest medical cost savings were US$17,903 per person after 35–56 days of survival. After 56 days of survival, the saved medical cost gradually decreased with the extension of the survival days. After almost one year of survival, the medical cost for patients receiving hospice care was not saved, but increased [32].

The cumulative cost savings found by Obermeyer et al. showed a U-shaped trend, which was mainly since terminally ill patients need more palliative care when they first receive hospice care or when they are closer to death [72]. However, a U-shaped trend was not observed in the present study. There are two plausible explanations for this difference: (1) the starting point of the medical cost savings was different after accepting hospice care, and (2) after accepting hospice care, the timepoint for saving most medical expenses was different.

The difference in the starting timepoint for saving medical expenses after accepting hospice care could be related to different exclusion conditions for research subjects. Hospice care patients in this study were not excluded from cancer-directed treatment (e.g., surgery, chemotherapy, and radiotherapy) due to the inclusion of hospice-shared care cases. Such cases were excluded by Obermeyer et al. The difference in the starting point of medical cost saving could also be related to high medical expenses accumulated before hospice care. The medical expenses of the patients in the hospice group in the study were higher than those in the traditional group 30 days before receiving hospice care, especially on the day of receiving hospice care, which was when the medical expenses of the hospice group reached its peak (Fig 6).

The difference in the time point for saving most medical expenses after accepting hospice care may have been related to the lower proportion of positive treatment in the hospice group compared with the traditional group. Additionally, the proportion of hospitalization for more than 14 days and in-hospital death decreased with the extension of survival days after receiving hospice care, although the proportion of use was higher than that of the traditional group (see Table 3). The fixed quota of the NHI system may also have been an influencing factor. The NHI system uses a fixed payment for medical expenses related to hospitalization and home-based hospice care [17]. Both the results of the present study and those of Obermeyer et al. found that early referral in hospice care can effectively save medical costs.

This study had some limitations. Cancer patients who were not covered by health insurance were not included as participants, and the related medical expenses for patients who had to upgrade to other wards at their own expense were not included in the analysis. However, because the coverage rate of Taiwan's health insurance was maintained at 96% from 2010 to

2013, the population used in this study was still representative of the target population. Due to the unavailability of cause-of-death documents in the health insurance database, the death date of this study was replaced by the death date of major injury files, and the information for the cause of death was not obtained. However, cancer patients constituted the majority of the patients applying for major injuries, accounting for about 50% of all major injuries.

Moreover, nearly 99% of cancer patients in Taiwan now have cards for major injuries [73]. Therefore, this study used the death dates of major injury and disease files to screen the cancer death population, and the results should have been similar to those of the real cancer death population. The date of hospice-shared care is unavailable from National Health Insurance Research Database (NHIRD), and the hospice-shared care was the service of palliative consultation in the present study. Thus, the medical health expense of hospice-shared care was calculated by the proportion of hospitalization day, and the cumulative cost saving of hospice-shared care cannot separate from all hospice patients.

In addition, the cultural considerations in Taiwan may impact the decision to enroll in hospice care. However, the religions, traditional philosophies and values, and family-related barriers in telling the truth could not be obtained from the national health insurance database. Future studies should investigate the relation of the cause of death and medical expenses in (EOL) and the questionnaires in which cultural consideration in EOL patient link to the database of national health insurance database adjusted for the influence factor of cultural consideration. Finally, future studies can attempt to calculate the cumulative cost savings of hospice-shared care.

## Conclusion

Hospice care not only reduces the proportion of terminal-stage patients receiving curative treatment and medical use but also reduces the cost of terminal-stage medical care. This study found that hospice care can effectively save on the cost of terminal-stage cancer medical care and that the earlier the hospice care is provided, the more cost-effective the care.

## Supporting information

**S1 Dataset. Minimal set of data.**
(XLSX)

## Acknowledgments

The authors would like to thank Ms. Siew Tzuh Tang for her expert advice. The study was based in part on data from the National Health Insurance Research Database (NHIRD) supported by the Bureau of National Health Insurance, Department of Health, and managed by the National Health Research Institutes (NHIRD-104-157). Editorial support was provided to the authors by Editage.

## Author Contributions

**Conceptualization:** Ya-Ting Huang, Ying-Wei Wang, Chou-Wen Chi, Wen-Yu Hu, Woung-Ru Tang.

**Data curation:** Ya-Ting Huang.

**Formal analysis:** Ya-Ting Huang, Rung Lin, Jr.

**Funding acquisition:** Ya-Ting Huang, Chih-Chung Shiao.

**Investigation:** Ya-Ting Huang, Woung-Ru Tang.

**Methodology:** Ya-Ting Huang, Ying-Wei Wang, Chou-Wen Chi, Wen-Yu Hu, Woung-Ru Tang.

**Project administration:** Ya-Ting Huang, Chih-Chung Shiao.

**Resources:** Ya-Ting Huang.

**Software:** Ya-Ting Huang.

**Supervision:** Ying-Wei Wang, Chou-Wen Chi, Wen-Yu Hu, Rung Lin, Jr, Woung-Ru Tang.

**Validation:** Woung-Ru Tang.

**Visualization:** Ying-Wei Wang, Woung-Ru Tang.

**Writing – original draft:** Ya-Ting Huang.

**Writing – review & editing:** Ya-Ting Huang, Woung-Ru Tang.

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
