## [Decision Letter · Decision Letter 0]

31 Oct 2019

PONE-D-19-21828

Differences in Medical Costs for End-of-Life Patients Receiving Traditional Care and Those Receiving Hospice Care: A Retrospective Study

PLOS ONE

Dear Mrs Huang,

Thank you for submitting your manuscript to PLOS ONE. After careful consideration, we feel that it has merit but does not fully meet PLOS ONE’s publication criteria as it currently stands. Therefore, we invite you to submit a revised version of the manuscript that addresses the points raised during the review process.

We would appreciate receiving your revised manuscript by Dec 15 2019 11:59PM. To enhance the reproducibility of your results, we recommend that if applicable you deposit your laboratory protocols in protocols.io, where a protocol can be assigned its own identifier (DOI) such that it can be cited independently in the future. For instructions see: http://journals.plos.org/plosone/s/submission-guidelines#loc-laboratory-protocols

We look forward to receiving your revised manuscript.

Kind regards,

Chun Chieh Yeh, M.D., Ph.D.

Academic Editor

PLOS ONE

Journal Requirements:

4. In the ethics statement in the manuscript and in the online submission form, please provide additional information about the patient records used in your retrospective study. Specifically, please ensure that you have discussed whether all data were fully anonymized before you accessed them and/or whether the IRB or ethics committee waived the requirement for informed consent. If patients provided informed written consent to have data from their medical records used in research, please include this information.

Additional Editor Comments (if provided):

Please revise the article in response to the reviewers' comments specifically. The whole article needs additional work by an English-language editor for improving its flow and style. If possible, please attached proof of professional English editing.

Reviewers' comments:

Reviewer's Responses to Questions

**Comments to the Author**

1. Is the manuscript technically sound, and do the data support the conclusions?

Reviewer #1: Partly

Reviewer #2: Yes

2. Has the statistical analysis been performed appropriately and rigorously? 

Reviewer #1: Yes

Reviewer #2: Yes

3. Have the authors made all data underlying the findings in their manuscript fully available?

Reviewer #1: No

Reviewer #2: No

4. Is the manuscript presented in an intelligible fashion and written in standard English?

Reviewer #1: Yes

Reviewer #2: Yes

5. Review Comments to the Author

Reviewer #1: The authors of this study have put together a well thought out analysis of cost associated with traditional versus hospice for patients in Taiwan with a diagnosis of terminal cancer. As was nicely stated in the introduction, the paper “compared the difference in medical expenses between traditional care patients and terminally ill cancer patients receiving hospice care using the same number of survival days to specifically evaluate the cost-effectiveness of hospice care and to provide a reference for all stakeholders.”

I think the analysis nicely provides a reference on cost for stakeholders. I am not sure, however, that “cost-effectiveness” is what is being measured. Cost-effectiveness examines the costs and outcomes of one or more interventions. So here, I think the intervention is hospice care and the health outcome is death. From the analysis as written, assessing cost-effectiveness would mean assuming that the outcome of death (matched so carefully between cases and controls) was inevitable. I am not sure that is the case – different therapies could have different effects on patients and their length of survival.

From the data source from the National Health Insurance database, is there information available on the causes of death? I saw that the cases were matched using cancer diagnosis, but did not see data on cause of death. If, for instance, one group had different causes of death than the other (i.e. related to but not definitively their specific cancer diagnosis), that could be an interesting finding. It could also have effects on cost-effectiveness.

The authors allude to cultural considerations in Taiwan that may impact the decision to enroll in hospice care. Do the authors think these are significant enough to make the groups different in their decision-making processes? In other words, do individuals who choose hospice care have different approaches to their health care that bring about the cost differences?

What do the authors think we should do with this data?

Reviewer #2: I think this is an exciting addition to the research on global hospice.

Major comments:

-The introduction needs additional work by an English-language editor for flow and style. The last two sentences of the first paragraph are editorial comments and appear out-of-place in a manuscript like this. Additional context about the status of and recent changes in hospice care provision in Taiwan should be in the introduction. I would recommend moving the discussion of the development of the hospice shared-care program from the discussion to the intro. An explicit description of the hospice shared-care population in the background would also provide needed context for analyzing the results. Overall, I think the background should include more of an "environmental scan" of end-of-life care in Taiwan: where does it occur, what are the barriers, and why is it important to analyze the cost factors addressed by this paper.

-Is it possible to separate out your results and images based on the various types of services that have been lumped together into "hospice" within the study? It would be particularly powerful in figure 5 to see shared-care hospice as a separate trend line. If this analysis isn't available, I strongly encourage a follow-up study for future publication.

Minor comments:

-In referring to "inpatient palliative care" for instance on page 8 when you describe the "hospice group," does this refer to the provision of inpatient expert palliative care team consultation on all those with serious illness (see the WHO definition of palliative care)? Or is this being used to describe those who received "hospice" end-of-life services in the hospital setting? If it is the latter, it needs to describe it that way rather than using the "inpatient palliative care" terminology so that it is more clear to an international audience what is being described. I would also point out that it appears to be described as "inpatient hospice care" in Table 1.

-In section 2, you discuss a 10-day interval; quick review of the accompanying citation (24570105) doesn't provide a 10-day interval, can there be more explanation of how this number was selected.

-Figure 2 needs better grayscale formatting (more like figures 4 or 5).

6. PLOS authors have the option to publish the peer review history of their article (what does this mean?). If published, this will include your full peer review and any attached files.

Reviewer #1: No

Reviewer #2: Yes: Kyle P. Edmonds, MD FAAHPM

---

## [Author Response · Author response to Decision Letter 0]

17 Dec 2019

Thank you for the generally positive comments on our manuscript. We also completely agree with your excellent comments. Our responses to your specific comments are listed in "rebuttal letter".

---

## [Decision Letter · Decision Letter 1]

3 Feb 2020

Differences in Medical Costs for End-of-Life Patients Receiving Traditional Care and Those Receiving Hospice Care: A Retrospective Study

PONE-D-19-21828R1

Dear Dr. Huang,

We are pleased to inform you that your manuscript has been judged scientifically suitable for publication and will be formally accepted for publication once it complies with all outstanding technical requirements.

With kind regards,

Chun Chieh Yeh, M.D., Ph.D.

Academic Editor

PLOS ONE

Additional Editor Comments (optional):

Congratulation to your great efforts. The content of manuscript could be accepted in its current version. However, if possible, it would be better to go through English editing to improve its readability.

Reviewers' comments:

Reviewer's Responses to Questions

**Comments to the Author**

1. If the authors have adequately addressed your comments raised in a previous round of review and you feel that this manuscript is now acceptable for publication, you may indicate that here to bypass the “Comments to the Author” section, enter your conflict of interest statement in the “Confidential to Editor” section, and submit your "Accept" recommendation.

Reviewer #2: All comments have been addressed

2. Is the manuscript technically sound, and do the data support the conclusions?

Reviewer #2: Yes

3. Has the statistical analysis been performed appropriately and rigorously? 

Reviewer #2: Yes

4. Have the authors made all data underlying the findings in their manuscript fully available?

Reviewer #2: No

5. Is the manuscript presented in an intelligible fashion and written in standard English?

Reviewer #2: Yes

6. Review Comments to the Author

Reviewer #2: (No Response)

7. PLOS authors have the option to publish the peer review history of their article (what does this mean?). If published, this will include your full peer review and any attached files.

Reviewer #2: Yes: Kyle P. Edmonds, MD FAAHPM

---

## [Editor Report · Acceptance letter]

10 Feb 2020

PONE-D-19-21828R1 

Differences in Medical Costs for End-of-Life Patients Receiving Traditional Care and Those Receiving Hospice Care: A Retrospective Study 

Dear Dr. Huang:

I am pleased to inform you that your manuscript has been deemed suitable for publication in PLOS ONE. Congratulations! Your manuscript is now with our production department. 

With kind regards,

on behalf of

Dr. Chun Chieh Yeh 

Academic Editor

PLOS ONE